# Suicide literacy and stigma among undergraduate students in Pokhara, Nepal: A cross-sectional study

Dhurba Khatri[1,2]*, Dipendra Kumar Yadav[1], Shishir Paudel[2], Sujan Poudel[3], Yamuna Chhetri[2], Sandeepa Karki[2]

**1** School of Health and Allied Sciences, Pokhara University, Kaski, Nepal, **2** Kathmandu Institute of Child Health, Kathmandu, Nepal, **3** HERD International, Lalitpur, Nepal

* kcdhurba06@gmail.com

## Abstract

### Background

Suicide is a major global public health concern, with stigma and low suicide literacy posing significant barriers to prevention and intervention. Despite increasing suicide rates and concerns about suicidal ideation, limited studies have assessed suicide stigma and literacy among young adults in Nepal.

### Objective

This study examines suicide literacy and stigma among undergraduate students and its associated factors.

### Methods

A cross-sectional study was conducted among 597 randomly selected undergraduate students in Pokhara Metropolitan City, Nepal, using the Literacy of Suicide Scale (LOSS-SF) and the Stigma of Suicide Scale (SOSS-SF). Descriptive statistics, independent t-tests, and ANOVA were used to analyze socio-demographic differences.

### Results

The mean suicide literacy score was 4.83±2.07, revealing significant knowledge gaps, particularly in suicide risk factors and misconceptions. Stigma levels were high, with "pathetic" (42.4%), "immoral" (41.7%), and "stupid" (40.5%) being commonly endorsed descriptors. The average scores were 23.15 (SD±13.12) for SOSS-stigma, 10.10 (SD±3.89) for glorification, and 13.12 (SD±4.28) for isolation/depression. Females had higher isolation ($p=0.001$) and literacy scores ($p=0.023$), while males exhibited greater glorification tendencies ($p<0.001$). Adults (≥20 years) had significantly higher stigma, isolation, and literacy scores than adolescents ($p<0.05$).

**Data availability statement:** All relevant data are within the paper and its Supporting Information files.

**Funding:** The study was carried out with financial support from Nepal Health Research Council as a post-graduation health research grant of the Year 2021. Award awarded to the Principal Investigator. The funders had no role in study design, data collection and analysis, decision to publish, or preparation of the manuscript.

**Competing interests:** The authors have declared that no competing interests exist.

Higher isolation and literacy scores were observed among students living with family (p < 0.05) and third-year students (p < 0.01). Maternal literacy and employment were associated with higher glorification scores (p < 0.05).

## Conclusion

These findings underscore the urgent need for suicide literacy programs and stigma-reduction initiatives in Nepalese educational institutions. Addressing misconceptions and fostering help-seeking behaviors through culturally sensitive interventions could enhance suicide prevention efforts.

## Introduction

Suicide is global public health crisis that is persistent and increasing in many parts of the world, leading to substantial economic, emotional and human cost [1,2]. It is one of the leading causes of death among young people and a significant contributor to the global burden of disease [3]. The global estimate suggests that each year 703, 000 people die by suicide marking it third leading cause of death among 15–29-year-olds [4]. The prevalence and risk of suicide differ significantly across demographics, emphasizing the need for targeted prevention efforts. It is not just a mental health issue but have a huge social impact affecting individuals, families, communities and societies and not just a single life lost [1].

Stigma refers to social isolation and being assigned an unfavorable social standing [5], which often manifests in emotional, cognitive, or behavioral forms [6]. Suicide stigma can be defined as unfavorable attitudes toward people with suicidal tendencies, which can also result in a variety of harmful effects [7]. Stigmatizing attitudes towards suicide promote social isolation among those experiencing mental burdens, potentially affecting adherence to treatment regimens and lowering self-esteem and perceptions of their need for support [8–13].

One critical factor influencing stigma and help-seeking behavior is mental health literacy, particularly literacy about suicidality [14]. Suicide literacy is a specific type of mental health literacy referring to the understanding of the different facets of suicidality including its warning signs/symptoms, causes of suicidality, risk factors, and proper treatment and prevention [7]. Higher levels of mental health literacy have been shown to significantly improve help-seeking behavior and reduce stigma [15–17]. A lack of adequate suicide literacy often exacerbates stigma, leading to negative outcomes for those at risk. The public understanding about suicide could directly influence the outcomes of individuals experiencing suicidal thoughts or behaviors [18].

The stigma associated with mental illness, particularly suicidality, has been considered as is one of the most critical obstacles to receiving professional psychological treatment [6]. Furthermore, low levels of suicide literacy are strongly linked to delays in identifying suicidal thoughts and providing timely intervention [7]. Evidences suggests that misunderstanding about risk factors, warning signs/symptoms and therapies related to mental issues could increase the likelihood of suicidal ideation and

stigmatization, which may further exacerbate suicidal thoughts and tendencies [19]. These findings emphasize the critical need to improve mental health literacy as a central strategy for suicide prevention.

Stigma and suicide literacy are interdependent factors that influence not only individual outcomes but also the broader development and implementation of effective suicide prevention programs. Addressing stigma and enhancing literacy about suicide are essential to creating a supportive environment that encourages help-seeking behaviors and reduces the societal burden of suicide. The National Mental Health Survey of Nepal 2020 has highlighted stigma as a major to accessing mental health services in the country [20]. Similarly, a systematic review of 57 studies assessing stigma toward mental health disorders in Nepal highlighted substantial gaps in implementing and evaluating stigma-reduction interventions [21]. However, while these studies have focused on mental health stigma, there is a lack of evidence specifically addressing suicidal stigma and suicide literacy.

Despite the importance of these factors, limited research in Nepal, particularly among youth populations, who are at a higher risk of suicide. This study aims to assess the status of suicide literacy and stigma among Nepalese college students. By identifying gaps in knowledge and attitudes, this research will inform the design of targeted interventions to improve mental health outcomes and reduce the stigma surrounding suicide in Nepal.

## Methods

### Study design and setting

A cross sectional study was conducted among at undergraduate students enrolled at different colleges in Pokhara Metropolitan City, located in the Kaski District, Gandaki Province, Nepal in from April 2022 to June 2022. The data were collected from April 2022 to June 2022. Pokhara metropolitan city is one largest metropolitan in Nepal and one important destination that attracts students from different part of the country for better education and career opportunities. The 2021 Census reported that the metropolitan accommodates 513,504 individuals aged 15–45 years to 272744 [22]. The included students from six major faculties available in Pokhara Metropolitan City to ensure broader academic representation: Engineering (including BSc Civil, Mechanical, Electrical, and Computer Engineering programs), Management (BBA, BBS, BBA-BI, BHM, BCA), Law (BALLB, LLB), Health and Allied Sciences (Medicine, Dental, Nursing, Public Health, Pharmacy, physiotherapy, optometry and Medical Laboratory Technology, optometry), Humanities (BDevS, Arts, Humanities, B.Ed), and Pure Sciences (BSc in Physics, Chemistry, Biology, Microbiology and Forestry).

### Sampling size and sampling procedure

All the students enrolled in undergraduate courses inside Pokhara Metropolitan city were eligible to be included as study population. The sample size for this study was was determined using the formula, $n = \frac{Z^2 p(1-p)}{d^2}$ Where, Z is the standard normal variate with value 1.96 at 95% confidence interval, p is the assumed prevalence (50%, as no prior studies had assessed suicidal stigma or suicide literacy, making 50% an optimal assumption), and d is the allowable margin of error (5%). Substituting these values, the calculated sample size was 384. A design effect of 1.5 was applied to account for clustering in the sampling design resulting in sample size of 576. Additionally, adjusting 5% non-response rate sample was optimized to 600 students. Approaching 600 students 597 students provided complete response, as this was above the required minimum sample size, the data from 597 student were included in this study.

The study used cluster sampling, where undergraduate programs offered at various institutions within Pokhara were considered as clusters, and the total number of students enrolled in each program was identified. The students enrolled in the first year were excluded, as they had just joined their courses and were not regular attendees. The required sample size for each program was determined by calculating the proportion of students in each program relative to the total student population of Pokhara Metropolitan City. The sample sizes for each program were adjusted to ensure a representative sample, accounting for variations across different fields of study. To select students within each program, a random

sampling method was employed. Each course, enrollment year (excluding first-year students), and class section was assigned a unique code. These codes were then randomly selected using the lottery method to ensure an unbiased and random selection process. The final number of students drawn from each program was determined based on their proportion within the total student population, ensuring a proportionate representation from each field of study. This method allowed for a diverse sample that accurately reflected the student population of Pokhara Metropolitan City.

## Data collection

The data were collected using a self-administered questionnaire in Nepali language, which was distributed to undergraduate students during a one-hour session organized with the assistance of school teachers. Researchers explained the study's purpose and guided students through the completion process, which took place in their classrooms within the allocated time frame. The questionnaire was divided into six sections where the first section consisted of background information about student's socio-demographic characteristics and the nature of their college. The second section was focus on mental health related characteristics such as history of mental illness, psychological counseling, family history of mental illness and suicide, along with Rosenberg Self-Esteem Scale (RSE) [23], which has been used to assess self-esteem among undergraduate students in Nepal [24–26], and also has been validated in Nepali language [27]. The third section consisted of the Multidimensional Scale of Perceived Social Support [28], validated in Nepali language [29] to assess students' level of perceived social support and questions on the students' tendency to seek assistance. The fourth section consisted of Suicidal Ideation Attributes Scale (SIDAS) [30]. Likewise, the fifth section consisted of Literacy of Suicide Scale short form (LOSS- SF) questionnaires [7,31,32], to assess the level of suicidal literacy among the students. And the last section consisted of Stigma of Suicide Scale (SOSS) questionnaires [33] to access the level of suicidal stigma among college students.

## Data analysis

The collected data were entered into EpiData version 3.1, with measures such as double entry, re-checking, and cleaning undertaken for quality assurance. Statistical analyses were conducted using Statistical Package for Social Science V.22 (IBM Corp. Armonk, NY, USA) [34] Descriptive statistics, including frequencies, percentages, means, and standard deviations, were used to summarize the data. The normality of the data was evaluated using the Shapiro-Wilk normality test, a common statistical tool used to assess the normality assumption of a dataset. As the data followed normal distribution independent student t-tests and ANOVA, were performed to examine differences in SOSS and LOSS scores based on participants' background characteristics. Post-hoc analysis was performed using the Turkey post-hoc test. Pearson correlation was performed between total scores of different Likert scale tools. Statistical significance was set at p-value < 0.05. The data used in this study is provided as a Supplementary File (S1 File).

## Outcome variables

The outcome variables were suicidal literacy assessed through LOSS and suicidal stigma assess through SOSS. The LOSS is a 12-item questionnaire used to assess knowledge about suicide, including suicide signs and symptoms (3 items), causes/nature (4 items), risk factors (3 items), and treatment and prevention (2 items) [7,33]. In SOSS items are scored on a 5-point Likert scale ranging from 1 = "strongly disagree" to 5 = "strongly agree". The questionnaire includes three subscales: stigmatizing attitudes towards suicide (8 items), attribution of isolation or depression on suicide (4 items), and normalization or glorification of suicide (4 items) [33]. To address potential language barriers, the research team consulted with bilingual mental health experts and conducted a pretest to ensure translation validity.

## Ethical considerations

Ethical approval for the study was obtained from the Institutional Review Committee of Pokhara University (Ref: 15/078/079), and permissions were secured from the participating academic institutions. Permission was also obtained

from the original developers to use the LOSS-SF and SOSS tools in this study. Data were collected only after obtaining the necessary approvals and securing written informed consent from all participants. Written parental consent was additionally obtained for the one participant who was under the age of 18. The parental consent form was provided to the student several days prior to data collection, and the signed consent form was collected before the student's participation. All participants were fully informed about the study's objectives, their participation was entirely voluntary, and confidentiality was strictly maintained. The collected data were used solely for research purposes, ensuring the privacy and anonymity of all respondents.

## Results

The study included a total of 597 participants with age ranging between 17–38 years and the mean age of 22.02 ± 2.27 years. The majority were female (64.3%), while males comprised 35.7% of the sample. Most participants were adults aged 20 years or older (90.3%), with only 9.7% classified as adolescents (≤19 years). Regarding academic year, the largest group of participants was in their second year of study (37.5%), followed by fourth-year (33%) and third-year students (29.5%). In terms of parental education, fathers more commonly attained secondary education (41.7%) or higher (22.4%), while mothers had lower rates of education at these levels (35.8% for secondary education and 12.1% for bachelor's degree or above). Conversely, mothers were more likely to be unemployed (43.9%) compared to fathers, who exhibited a significantly higher employment rate (91.1%) (Table 1).

Out of total participants, only 3% of participants reported having a chronic disease, and 4.4% had a history of mental illness diagnosed by a psychiatrist. Among those reporting mental illness (n = 26), anxiety and depression were the most common diagnoses (30.8% each), followed by overthinking (26.7%). Rare conditions included insomnia, attention deficit disorder, and bipolar disorder (3.8% each). A minority of participants (2.3%) reported a family history of mental illness, and 2.8% reported a history of suicide in their family within the past three generations. The majority of participants (95.1%) reported high perceived support from family, while 4.9% experienced moderate support. Similarly, high support from friends was common (73.9%), followed by moderate (23.5%) and low support (2.7%). Perceived support from others was also largely high (75.2%), with 18.3% reporting moderate support and 6.5% reporting low support. Self-esteem levels were generally normal (75.4%), though a significant proportion of participants (20.4%) reported low self-esteem (Table 2).

The mean score of LOSS-SF was 4.83 ± 2.07. The highest correct response rate was for the item "Seeing a psychiatrist or psychologist can help prevent someone from suicide" (82.2%), while the lowest was for "If assessed by a psychiatrist, everyone who suicides would be diagnosed as depressed" (15.6%) (Table 3).

In regards to Stigma of Suicide Scale, the participants scored an average of 23.15 (SD ± 13.12) on the SOSS stigma subscale, 10.10 (SD ± 3.89) on glorification, and 13.12 (SD ± 4.28) on isolation/depression. Among the stigma-related characteristics, "pathetic" (42.4%), "immoral" (41.7%), and "stupid" (40.5%) were some of the most commonly agreed upon descriptors, each with mean scores near 3. Similarly, 39.4% of respondents viewed suicide as "irresponsible", with a mean of 2.96 (SD ± 1.34). The perception of suicide as "cowardly" had the highest agreement (48.7%) among with a mean score of 2.79 (SD ± 1.35). Interestingly, descriptors associated with glorification, such as "strong" (62.3%), "brave" (63.3%), and "novel" (59.6%), showed higher levels of agreement (Table 4).

Females exhibited higher mean scores in SOSS-Isolation (12.53 ± 4.14, p = 0.001) and LOSS-SF Literacy (4.97 ± 2.12, p = 0.023), whereas males scored higher in SOSS-Glorification (10.82 ± 4.48, p < 0.001). Adults (≥20 years) reported significantly greater stigma (23.35 ± 6.06, p = 0.016), isolation (13.30 ± 4.29, p = 0.002), and literacy (4.89 ± 2.07, p = 0.030) compared to adolescents (≤19 years). Those living with family had higher isolation (13.32 ± 4.38, p = 0.047) and literacy scores (4.94 ± 2.12, p = 0.022) than those living without family. Third-year bachelor's students exhibited significantly higher scores in isolation (13.94 ± 4.04, p = 0.007), glorification (9.96 ± 3.77, p < 0.001), and literacy (5.25 ± 2.05, p = 0.006) compared to other academic levels. Maternal literacy was associated with higher glorification scores (10.21 ± 3.90 vs. 8.70 ± 3.55, p = 0.015), while maternal employment showed marginally higher glorification scores (10.38 ± 4.07 vs. 9.74 ± 3.63, p = 0.047) (Table 5).

**Table 1. Socio-demographic characteristics of the participants.**

| Variables | n (%) |
|---|---|
| **Gender** | |
| Male | 213 (35.7) |
| Female | 384 (64.3) |
| **Age** | |
| Adolescent (≤19 years) | 58 (9.7) |
| Adult (≥20 years) | 539 (90.3) |
| **Faculty** | |
| Engineering | 113 (18.9) |
| Law | 26 (4.4) |
| Management | 252 (42.2) |
| Health and Allied Sciences | 76(12.7) |
| Education | 25(4.2) |
| Humanities | 39(6.5) |
| Sciences | 66 (11.1) |
| **Academic year** | |
| 2nd year bachelor | 224 (37.5) |
| 3rd year bachelor | 176 (29.5) |
| 4th year bachelor | 197 (33.0) |
| **Ethnicity** | |
| Upper Caste | 406 (68.0) |
| Janajati | 139 (23.3) |
| Dalit | 52 (8.7) |
| **Religion** | |
| Hindu | 539 (90.3) |
| Non-hindu | 58 (9.7) |
| **Type of college** | |
| Private | 310 (51.9) |
| Public | 287 (48.1) |
| **Education level of father** | |
| Illiterate | 17 (2.8) |
| Informal schooling | 27 (4.5) |
| Basic (1–8 Class) | 170 (28.5) |
| Secondary (9–12 Class) | 249 (41.7) |
| Bachelor and above | 134 (22.4) |
| **Education level of mother** | |
| Illiterate | 44 (7.4) |
| Informal schooling | 57 (9.5) |
| Basic (1–8 Class) | 210 (35.2) |
| Secondary (9–12 Class) | 214 (35.8) |
| Bachelor and above | 72 (12.1) |
| **Mothers employment status** | |
| Yes | 335 (56.1) |
| No | 262 (43.9) |
| **Fathers employment status** | |
| Yes | 544 (91.1) |
| No | 53 (8.9) |

Engineering included Civil, IT, Electrical; Law included BALLB and LLB; Management included BBA, BBS, BCA, Health and Allied Sciences included MBBS, BPH, Nursing and Pharmacy; Education included B.Ed; Humanities included Developmental studies and Arts; Sciences included BSc Physics, Chemistry, Microbiology.

**Table 2. Psychosocial and health characteristics of participants.**

| Variables | n (%) |
|---|---|
| **Chronic Disease** | |
| Yes | 18 (3.0) |
| No | 579 (97.0) |
| **History of Mental Illness** | |
| Yes | 26 (4.4) |
| No | 571 (95.6) |
| **Family History of mental illness** | |
| Yes | 14 (2.3) |
| No | 583 (97.7) |
| **Family History of Suicide** | |
| Yes | 17 (2.8) |
| No | 580 (97.2) |
| **Taken any kind of Psychological counseling** | |
| Yes | 24 (4.0) |
| No | 573 (96.0) |
| **Perceived Support from Family** | |
| Moderate support | 29 (4.9) |
| High support | 568 (95.1) |
| **Perceived Support Friends** | |
| Low support | 16 (2.7) |
| Moderate support | 140 (23.5) |
| High support | 441 (73.9) |
| **Perceived Support Others** | |
| Low support | 39 (6.5) |
| Moderate support | 109 (18.3) |
| High support | 449 (75.2) |
| **Self-esteem level** | |
| Low self-esteem | 122 (20.4) |
| Normal self-esteem | 450 (75.4) |
| Higher Self Esteem | 25 (4.2) |

Individuals with moderate social support had significantly higher scores on LOSS-SF Literacy ($p = 0.029$), indicating lower suicide literacy in this group. Additionally, participants with low self-esteem exhibited significantly higher SOSS-Isolation scores ($p = 0.024$), reflecting a greater sense of isolation related to suicide stigma. No significant differences were observed across other variables, including mental illness, family history of mental illness or suicide, and perceived social support from friends or others (Table 6). The additional results of the multiple linear regression analyses, including full coefficient tables for all models (SOSS-Stigma, SOSS-Depression, SOSS-Glorification and LOSS-SF) are provided in Supplementary File (S1 Table).

The correlation analysis revealed a strong positive association between SOSS Stigma and SOSS Depression ($r = 0.647$, $p < 0.01$), indicating that higher stigma is linked to greater depression-related stigma. SOSS Glorification showed a weak inverse relationship with SOSS Depression ($r = -0.137$, $p < 0.01$). LOSS total score did not exhibit significant correlations with other variables, suggesting minimal association. The MSPSS family support score was

**Table 3. Participant response on Literacy of Suicide Scale short form.**

| Item | Dimension | n(%) |
|---|---|---|
| 1. If assessed by a psychiatrist, everyone who suicides would be diagnosed as depressed (F) | Cause/Nature | 93 (15.6) |
| 2. Talking about suicide always increases the risk of suicide (F) | Cause/Nature | 307 (51.4) |
| 3. Very few people have thoughts about suicide (F) | Cause/nature | 166 (27.8) |
| 4. A suicidal person will always be suicidal and entertain thoughts of suicide (F) | Cause/Nature | 206 (34.5) |
| 5. Seeing a psychiatrist or psychologist can help prevent someone from suicide (T) | Treatment/Prevention | 491 (82.2) |
| 6. People who have thoughts about suicide should not tell others about it (F) | Treatment/prevention | 320 (53.6) |
| 7. Most people who suicide are psychotic (F) | RiskFactor | 122 (20.4) |
| 8. There is a strong relationship between alcoholism and suicide (T) | RiskFactor | 251 (42.0) |
| 9. Men are more likely to suicide than women (T) | RiskFactor | 191 (32.0) |
| 10. People who talk about suicide rarely kill themselves (F) | Sign/Symptom | 138 (23.1) |
| 11. People who want to attempt suicide can change their mind quickly (T) | Sign/Symptom | 268 (44.9) |
| 12. Not all people who attempt suicide plan their attempt in advance (T) | SingSymptom | 334 (55.9) |
| Over all Mean ± SD = 4.83 ± 2.07 | | |

**Table 4. Responses Stigma of Suicide Scale.**

| Characteristics | Agreement* n (%) | Mean (SD) |
|---|---|---|
| Pathetic | 253 (42.4) | 2.82 ± 1.27 |
| Shallow | 207 (34.7) | 2.99 ± 1.24 |
| Immoral | 249 (41.7) | 2.84 ± 1.23 |
| An embarrassment | 172 (28.8) | 3.18 ± 1.23 |
| Irresponsible | 235 (39.4) | 2.96 ± 1.34 |
| Stupid | 242 (40.5) | 3.00 ± 1.43 |
| Cowardly | 291 (48.7) | 2.79 ± 1.35 |
| Vengeful | 311 (52.1) | 2.59 ± 1.27 |
| Lonely | 169 (28.3) | 3.44 ± 1.42 |
| Isolated | 174 (29.1) | 3.20 ± 1.23 |
| Lost | 188 (31.5) | 3.22 ± 1.36 |
| Disconnected | 171 (28.6) | 3.26 ± 1.29 |
| Strong | 372 (62.3) | 2.43 ± 1.32 |
| Brave | 378 (63.3) | 2.46 ± 1.44 |
| Novel | 356 (59.6) | 2.38 ± 1.27 |
| Dedicate | 258 (43.2) | 2.84 ± 1.29 |
| SOSS-SF stigma | | 23.15 ± 13.12 |
| SOSS-SF glorification | | 10.10 ± 3.89 |
| SOSS-SF Isolation/ Depression | | 13.12 ± 4.28 |

*Agreement refers to participants agreeing or strongly agreeing with the respective statement

weak positive correlated with Self-Esteem (r = 0.211, p < 0.01). Similarly, there was a weak positive correlation between MSPSS Others and MSPSS Family (r = 0.315, p < 0.01) as well as between MSPSS Friend and MSPSS Family (r = 0.392, p < 0.01). These findings underscore the interplay between stigma, depression, and the protective role of social support in mental health (Table 7).

**Table 5. Comparison of mean scores with Socio-demographic variable and SOSS and LOSS.**

| Variables | SOSS Stigma | | SOSS Isolation | | SOSS Glorification | | LOSS-SF Literacy | |
|---|---|---|---|---|---|---|---|---|
| | Mean±SD | p-value | Mean±SD | p-value | Mean±SD | p-value | Mean±SD | p-value |
| **Gender#** | | | | | | | | |
| Male | 22.57±6.50 | 0.594 | 12.36±4.44 | 0.001* | 10.82±4.48 | <0.001* | 4.57±1.95 | 0.023* |
| Female | 23.25±5.99 | | 12.53±4.14 | | 9.7±3.47 | | 4.97±2.12 | |
| **Age#** | | | | | | | | |
| Adolescent (≤19 years) | 21.31±6.87 | 0.016* | 11.44±3.84 | 0.002* | 9.15±3.41 | 0.051 | 4.27±1.92 | 0.030* |
| Adult (≥20 years) | 23.35±6.06 | | 13.30±4.29 | | | | 4.89±2.07 | |
| **Marital Status#** | | | | | | | | |
| Married | 22.76±6.96 | 0.668 | 13.60±4.86 | 0.443 | 8.41±3.58 | 0.003* | 5.32±1.94 | 0.108 |
| Unmarried | 23.18±6.11 | | 13.08±4.24 | | 10.23±3.89 | | 4.79±2.07 | |
| **Accommodation Type#** | | | | | | | | |
| Family | 23.38±6.27 | 0.109 | 13.32±4.38 | 0.047* | 10.06±3.83 | 0.697 | 4.94±2.12 | 0.022* |
| Without family | 22.44±5.83 | | 12.51±3.90 | | 10.21±4.09 | | 4.49±1.85 | |
| **Faculty +** | | | | | | | | |
| Engineering [a] | 21.92±6.84 | <0.001* a<d; f<b, c,d | 11.95±4.67 | <0.001* a<c,d; f<c,d,e, g | 10.71±4.63 | 0.101 | 4.09±1.88 | <0.001* a<b,c,d; f<d |
| Law [b] | 24.88±5.65 | | 13.19±4.21 | | 9.23±3.16 | | 5.61±2.22 | |
| Management [c] | 23.50±5.70 | | 13.46±3.95 | | 10.27±3.74 | | 5.09±2.11 | |
| Health and allied Sciences [d] | 25.25±5.46 | | 14.71±3.94 | | 9.07±3.29 | | 5.44±1.86 | |
| Education [e] | 23.08±7.56 | | 13.48±4.27 | | 10.52±3.41 | | 4.36±1.89 | |
| Humanities [f] | 19.64±6.55 | | 10.10±3.98 | | 9.71±4.29 | | 4.05±2.07 | |
| Sciences [g] | 22.95±5.78 | | 13.59±4.25 | | 9.96±3.79 | | 4.74±1.98 | |
| **Education level +** | | | | | | | | |
| 2nd year bachelor [a] | 22.88±6.49 | 0.104 | 12.60±4.22 | 0.007* a<b | 9.44±3.57 | <0.001* c>a,b | 4.66±2.05 | 0.006* b>a,c |
| 3rd year bachelor [b] | 23.98±5.50 | | 13.94±4.04 | | 9.96±3.77 | | 5.25±2.05 | |
| 4th year bachelor [c] | 22.72±6.32 | | 12.96±4.46 | | 10.97±4.20 | | 4.65±2.05 | |
| **Ethnicity +** | | | | | | | | |
| Upper Caste | 23.00±5.99 | 0.457 | 13.02±4.31 | 0.325 | 10.14±3.86 | 0.807 | 4.85±2.12 | 0.892 |
| Janajati | 23.72±6.49 | | 13.56±4.18 | | 10.10±3.96 | | 4.76±2.03 | |
| Dalit | 22.84±6.66 | | 12.69±4.32 | | 9.76±4.03 | | 4.84±1.76 | |
| **Religion#** | | | | | | | | |
| Hindu | 23.22±6.16 | 0.419 | 13.08±4.31 | 0.519 | 10.16±3.91 | 0.229 | 4.85±2.09 | 0.528 |
| Non-hindu | 22.53±6.34 | | 13.46±4.02 | | 9.51±3.69 | | 4.67±1.89 | |
| **Type of College#** | | | | | | | | |
| Private | 23.05±6.16 | 0.664 | 13.22±4.56 | 0.546 | 10.33±4.16 | 0.126 | 4.90±2.06 | 0.432 |
| Public | 23.27±6.19 | | 13.01±3.96 | | 9.85±3.57 | | 4.76±2.08 | |
| **Education level of Father #** | | | | | | | | |
| Illiterate | 23.23 ±5.2 | 0.958 | 12.82 ±2.9 | 0.772 | 9.05 ±3.17 | 0.263 | 4.29±2.11 | 0.275 |
| Literate | 23.15 ±6.2 | | 13.12±4.32 | | 10.13±3.91 | | 4.85±2.07 | |
| **Education level of Mother#** | | | | | | | | |
| Illiterate | 23.27±6.68 | 0.898 | 13.29±3.94 | 0.779 | 8.7 ±3.55 | 0.015* | 4.75±1.98 | 0.776 |
| Literate | 23.14±6.14 | | 13.10±4.31 | | 10.21±3.90 | | 4.84±2.08 | |
| **Mothers Employment Status#** | | | | | | | | |
| Yes | 23.01±6.38 | 0.516 | 13.03±4.23 | 0.598 | 10.38±4.07 | 0.047* | 4.74±2.04 | 0.218 |
| No | 23.34±5.90 | | 13.22±4.35 | | 9.74 ±3.63 | | 4.95±2.10 | |

*(Continued)*

**Table 5.** (Continued)

| Variables | SOSS Stigma | | SOSS Isolation | | SOSS Glorification | | LOSS-SF Literacy | |
|---|---|---|---|---|---|---|---|---|
| | Mean±SD | p-value | Mean±SD | p-value | Mean±SD | p-value | Mean±SD | p-value |
| **Fathers Employment Status#** | | | | | | | | |
| Yes | 23.26±6.09 | 0.182 | 13.12±4.25 | 0.883 | 10.08±3.84 | 0.752 | 4.87±2.07 | 0.139 |
| No | 22.07±6.95 | | 13.03±4.61 | | 10.26±4.46 | | 4.43±1.98 | |

#Independent student t-test; +One way ANOVA;

*Statistical significance p<0.05

**Table 6. Comparison of mean scores with mental health related variable and SOSS and LOSS.**

| Variables | SOSS Stigma | | SOSS-SF Isolation | | SOSS- SF Glorification | | LOSS-SF Literacy | |
|---|---|---|---|---|---|---|---|---|
| | Mean±SD | p-value | Mean±SD | p-value | Mean±SD | p-value | Mean±SD | p-value |
| **Mental Illness #** | | | | | | | | |
| Yes | 22.15±5.67 | 0.397 | 13.34±4.45 | 0.784 | 11.00±3.63 | 0.230 | 4.83±2.26 | 0.944 |
| No | 23.20±6.19 | | 13.11±4.28 | | 10.06±3.90 | | 4.83±2.06 | |
| **Family History of mental illness #** | | | | | | | | |
| Yes | 25.00±6.44 | 0.259 | 14.50±4.45 | 0.223 | 10.28±5.26 | 0.859 | 5.28±1.85 | 0.412 |
| No | 23.11±6.16 | | 13.08±4.28 | | 10.09±3.86 | | 4.82±2.07 | |
| **Family History of Suicide #** | | | | | | | | |
| Yes | 25.70 ±6.36 | 0.084 | 13.76 ±3.81 | 0.530 | 10.29 ±2.14 | 0.837 | 5.41±1.87 | 0.245 |
| No | 13.76 ±3.81 | | 13.10 ±4.30 | | 10.09 ±3.93 | | 4.81±2.07 | |
| **Perceived Social Support from friend +** | | | | | | | | |
| Low social support | 24.75±5.05 | 0.411 | 13.25±3.51 | 0.917 | 9.56±3.14 | 0.399 | 5.75±2.26 | 0.190 |
| Moderate social support | 22.73±6.61 | | 13.24±4.57 | | 10.47±4.15 | | 4.75±2.28 | |
| High social support | 23.23±6.17 | | 13.07±4.22 | | 10.00±3.83 | | 4.82±1.99 | |
| **Perceived Social Support from family #** | | | | | | | | |
| Moderate social support | 23.96±5.31 | 0.412 | 14.24±3.34 | 0.149 | 10.89±3.58 | 0.261 | 5.65±2.48 | 0.029* |
| High social support | 23.11±6.21 | | 13.06±4.32 | | 10.06±3.91 | | 4.79±2.04 | |
| **Perceived Social Support from others+** | | | | | | | | |
| Low social support | 21.92±6.11 | 0.428 | 12.20±4.02 | 0.132 | 9.71±3.45 | 0.767 | 5.12±2.28 | 0.646 |
| Moderate social support | 23.33±5.76 | | 13.72±3.95 | | 10.24±3.79 | | 4.85±2.13 | |
| High social support | 23.22±6.17 | | 13.05±4.37 | | 10.10±3.96 | | 4.80±2.03 | |
| **Self-esteem level +** | | | | | | | | |
| Low self-esteem | 23.32±6.26 | 0.428 | 13.90±4.20 | 0.024* | 9.78±3.67 | 0.376 | 4.81±2.09 | 0.485 |
| Normal self-esteem | 23.19±6.09 | | 12.98±4.25 | | 10.22±3.92 | | 4.86±2.08 | |
| Higher Self Esteem | 21.60±7.26 | | 11.68±7.26 | | 9.44±4.45 | | 4.36±1.57 | |

#Independent student t-test; +One way ANOVA;

*Statistical significance p<0.05

## Discussion

This study aimed to assess the levels of suicidal literacy and stigma among undergraduate students in Pokhara Metropolitan City, Nepal, and identify associated factors influencing these constructs. The findings revealed that suicide literacy among students was relatively low, with a mean score of 4.83 out of 12, indicating that nearly half of the participants lacked adequate knowledge about suicide-related concepts. Comparatively, this level of literacy was similar to findings

**Table 7. Correlation Analysis among Stigma, LOSS, SIDAS, Self-Esteem, and MSPSS Subscales.**

Correlations

| | Stigma | | | LOSS | SIDAS | Self Esteem | MSPSS Others | MSPSS Family | MSPSS Friend |
|---|---|---|---|---|---|---|---|---|---|
| | SOSS Stigma | SOSS Depression | SOSS Glorification | | | | | | |
| SOSS Stigma Average | 1 | – | – | – | – | – | – | – | – |
| SOSS Depression | .647** | 1 | – | – | – | – | – | – | – |
| SOSS Glorification | 0.024 | −.137** | 1 | – | – | – | – | – | – |
| LOSS Total Score | −0.018 | 0.045 | −0.066 | 1 | – | – | – | – | – |
| Total SIDAS Score | −0.032 | −0.023 | 0.020 | 0.042 | 1 | – | – | – | – |
| Self-Esteem Scale | −0.025 | −0.075 | −0.013 | −0.027 | −0.078 | 1 | – | – | – |
| MSPSS Others | 0.009 | −0.016 | −0.016 | −0.040 | −0.025 | .138** | 1 | – | – |
| MSPSS Family | −0.038 | −.126** | −0.068 | −.181** | −0.073 | .211** | .315** | 1 | – |
| MSPSS Friend | −0.064 | −.094* | −0.021 | −0.071 | −.084* | .199** | .747** | .392** | 1 |

**Correlation is significant at the 0.01 level (2-tailed). *Correlation is significant at the 0.05 level (2-tailed).

from Bangladesh (4.27) [35] and China (5.83) [36] but lower than Turkey (9.96) [37] and Australia (17.0–20.4) [31,38]. Furthermore, a recent study conducted among Nepalese medical and nursing students reported a higher mean literacy score (6.36 ± 1.92) when using the validated Nepali version of the Literacy of Suicide Scale (LOSS-SF-Nep) [39], suggesting lower suicidal literacy even among medical and nursing students. Likewise, a study conducted in Bangladesh found that the mean score of the LOSS scale was 4.27 ± 1.99, with literacy significantly higher among females, medical students, and individuals with a history of suicidal attempts, while stigma was lower among females and those with a history of past attempts [35]. Similarly, research on Australian medical students demonstrated that students who normalized suicide had significantly lower intentions of seeking help for suicidal thoughts [38]. Another study among Arab youth revealed low suicide literacy (mean LOSS score = 3.82 ± 2.13), high levels of stigma (mean stigma subscale score = 3.24 ± 0.74), and strong perceptions of suicide as an isolating experience (mean isolation subscale score = 3.63 ± 0.74). Interestingly, attitudes toward seeking professional psychological help were positively correlated with suicide literacy and negatively correlated with both stigma and glorification [40]. These disparities suggest potential differences in mental health education, cultural perspectives on suicide, and accessibility of mental health resources across countries. These findings emphasize the critical need to address stigma and enhance suicide literacy through targeted educational interventions.

In this study participants demonstrated stronger knowledge about suicide prevention and treatment but had significant misconceptions regarding the causes and symptoms of suicidality. Specifically, while 82.2% correctly identified that seeing a psychiatrist or psychologist could help prevent suicide, only 15.6% understood that not all individuals who die by suicide would be diagnosed as depressed. These misconceptions align with previous studies indicating that poor awareness of risk factors and symptoms contributes to stigma and delayed intervention [35,37]. This highlights the need for targeted psychoeducation programs integrated into academic curricula to address these gaps in suicide literacy.

In this study, we did not exclude students based on their specific academic discipline, including those enrolled in health-related fields such as Public Health, Pharmacy, and Nursing. It is recognized that medical and allied health students are likely to have greater exposure to suicide-related content through their curriculum, potentially influencing their suicide literacy and stigma levels. However, evidence from Nepal presents mixed findings. Shah et al. (2022) reported a high total mean suicide literacy score (LOSS) of 13.07 (SD = 3.65) among doctors and nurses [41], whereas Gupta et al. (2023) found a mean score of 6.36 ± 1.92 using the validated Nepali version of LOSS-SF, with literacy rates ranging from

37.9% to 89.7% [39]. In the present study, students from allied health sciences demonstrated relatively higher suicide literacy scores (mean 5.44 ± 1.86) compared to students from other faculties such as Engineering (mean 4.09 ± 1.88) and Pure Sciences (mean 4.05 ± 2.07). This suggests that academic exposure might influence suicide literacy to some extent, although literacy gaps persisted across all disciplines.

This study also assessed the stigma associated with suicide, finding that negative perceptions, such as labeling suicide as "pathetic," "immoral," and "stupid," were prevalent. The mean stigma subscale score was 23.15 ± 13.12, consistent with studies in Turkey [37] and additionally, the study among Arab youth found that six out of the eight stigma-related items were endorsed by more than 50% of the sample, indicating substantial stigma toward suicidal individuals [40]. However, there was also a considerable level of glorification of suicide, with descriptors such as "brave" (63.3%) and "strong" (62.3%) receiving notable agreement while mean for SOSS-SF glorification subscale was 10.10 ± 3.89. Similar glorification trends have been observed in studies among Chinese and Australian students, where individuals with depression and suicidal thoughts were more likely to romanticize suicide [42]. These findings underscore the dual challenge of addressing both stigmatization and glorification, as both extremes can negatively influence help-seeking behaviors and suicide prevention efforts.

In terms of demographic variations, females exhibited significantly higher suicide literacy scores compared to males, whereas males had greater glorification tendencies (p < 0.001). This is consistent with studies from Australia and Germany, where males were found to hold stronger stigmatizing attitudes but lower depression-related attributions [38,43,44]. A study in Bangladesh also reported that stigma was significantly lower among females and those with a history of suicidal attempts [35]. Additionally, third- and fourth-year students scored higher in both the isolation and glorification subscales as compared to second year students, suggesting that academic stress and uncertainty about the future may influence suicide perceptions. These findings warrant targeted mental health interventions tailored to different academic years, ensuring that students receive adequate psychological support as they progress through their education.

The study found that students with lower self-esteem had significantly higher isolation scores, indicating that self-worth may play a critical role in how suicide is perceived. Correlation analysis further revealed that lower suicide literacy was associated with low family support (p = 0.005), while higher isolation scores were significantly linked with low self-esteem (p = 0.024). These findings highlight the critical role of social support and self-esteem in influencing both suicide literacy and stigma. Poor self-esteem could contribute to feelings of alienation, depression, and stress, which may increase the risk of both suicide and social isolation [24,45]. These findings underscore the complex interplay between self-esteem and mental health outcomes, emphasizing the importance of targeted interventions to address self-esteem issues as part of suicide prevention strategies. Additionally, the study revealed that strong social support was positively associated with higher suicide literacy and lower stigma. This aligns with international research suggesting that robust social networks can serve as protective factors, fostering mental health awareness and reducing suicide risk [46,47]. Enhancing peer support programs and family engagement initiatives could therefore be instrumental in improving suicide literacy and reducing stigma within communities. Notably, a significant association was observed between higher suicide literacy and stronger perceived social support from family (p = 0.022), reinforcing the role of supportive family environments in promoting mental health awareness. In line with these findings, a study among Arab youth demonstrated that attitudes toward seeking professional psychological help were positively correlated with suicide literacy and negatively correlated with stigma and glorification [40,48]. This suggests that fostering positive attitudes toward professional mental health services could be a key factor in improving suicide literacy and reducing stigma. Interestingly, no significant correlation was found between suicide literacy and personal or family history of mental illness, nor previous psychiatric consultations. This observation mirrors findings from studies conducted in China and Turkey [36,37]. These results highlight a critical gap, suggesting that formal mental health experiences do not necessarily enhance suicide literacy. Addressing this gap may require the implementation of structured mental health education programs that go beyond individual experiences to provide comprehensive, evidence-based knowledge about suicide and mental health.

The findings of this study have important implications for suicide prevention strategies in Nepal. Given the low literacy and high stigma scores, public health campaigns should focus on dispelling myths, normalizing discussions about mental health, and promoting help-seeking behaviors. Educational institutions can play a crucial role in implementing evidence-based mental health literacy programs and fostering an environment where students feel safe discussing mental health concerns. Additionally, interventions should be tailored to address gender differences in stigma and literacy, ensuring that male students receive targeted de-stigmatization efforts while enhancing female students' knowledge retention through interactive learning approaches.

While this study provides valuable insights into suicide stigma and literacy among Nepalese college students, several limitations should be considered when interpreting the findings. The sample was drawn from a specific metropolitan area, which may not fully capture the diverse experiences of students from different geographical and socio-economic backgrounds. Future studies could address this by including participants from rural and semi-urban settings to provide a more comprehensive understanding of suicide literacy and stigma across different contexts. Another potential limitation of the study could be the instruments used for data collection. At the time of the study (2022), the validated Nepali versions of the Literacy of Suicide Scale (LOSS-SF-Nep) and Stigma of Suicide Scale (SOSS-SF-Nep) were not yet published. Therefore, we utilized an expert-translated version, ensuring translation validity through expert consultation and field pretesting. However, the availability of validated Nepali versions by Gupta et al. (2023) marks a crucial advancement [39]. Future researchers are strongly encouraged to use the standardized validated Nepali versions to enhance the reliability and cultural appropriateness of assessments. Additionally, while this study examined various socio-demographic and psychological factors, other potential influences such as religious beliefs, cultural perceptions, and media exposure and its relation with suicidal literacy and stigma were not explored. Future research could benefit from qualitative approaches to better understand the nuanced factors that shape suicide literacy and stigma. Furthermore, longitudinal studies would be valuable in assessing how suicide literacy and stigma evolve over time and in response to targeted interventions. Expanding research to include intervention-based studies could provide insights into effective strategies for improving mental health awareness and reducing stigma related to suicide.

## Conclusion

This study highlights considerable gaps in suicide literacy and a notable presence of stigmatizing attitudes among undergraduate students in Pokhara, Nepal. Associations between suicide literacy and demographic variables such as gender, academic year, and age suggest that these factors may influence students' knowledge and perceptions. These findings support the need for targeted educational interventions that address specific misconceptions and stigma among Nepalese youth.

## Supporting information

**S1 File. Dataset.**
(XLSX)

**S2 File. Multiple linear regression analyses, including full coefficient for all models (SOSS-Stigma, SOSS-Depression, SOSS-Glorification and LOSS-SF).**
(DOCX)

**S1 Table. Multiple linear regression analysis of factors associated with suicide stigma (SOSS Stigma Average) among undergraduate students in Pokhara, Nepal.**
(DOCX)

## Acknowledgments

We would like to thank all the students who participated in this study and provided their valuable time and information.

## Author contributions

**Conceptualization:** Dhurba Khatri.

**Data curation:** Dhurba Khatri, Yamuna Chhetri.

**Formal analysis:** Dhurba Khatri, Shishir Paudel.

**Funding acquisition:** Dhurba Khatri.

**Investigation:** Dhurba Khatri.

**Methodology:** Dhurba Khatri, Dipendra Kumar Yadav, Shishir Paudel, Sujan Poudel.

**Project administration:** Dhurba Khatri.

**Resources:** Dhurba Khatri.

**Supervision:** Dipendra Kumar Yadav.

**Validation:** Dhurba Khatri, Dipendra Kumar Yadav.

**Visualization:** Dhurba Khatri, Shishir Paudel, Sujan Poudel.

**Writing – original draft:** Dhurba Khatri, Shishir Paudel.

**Writing – review & editing:** Sujan Poudel, Yamuna Chhetri, Sandeepa Karki.

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
