## [Decision Letter · Decision Letter 0]

8 Apr 2025

PONE-D-25-08118

Suicide Literacy and Stigma among Undergraduate Students in Pokhara, Nepal: A Cross-Sectional Study

PLOS ONE

Dear Dr. Khatri,

Thank you for submitting your manuscript to PLOS ONE. After careful consideration, we feel that it has merit but does not fully meet PLOS ONE’s publication criteria as it currently stands. Therefore, we invite you to submit a revised version of the manuscript that addresses the points raised during the review process.

Please respond to all comments from both the reviewers carefully. 

We look forward to receiving your revised manuscript.

Kind regards,

Saraswati Dhungana, MD

Academic Editor

PLOS ONE

 [The study was carried out with financial support from Nepal Health Research Council as a post-graduation health research grant of the Year 2021.  Award awarded to the Principal Investigator.]. 

4. In the online submission form, you indicated that [The data generated during the study are available from the corresponding author upon reasonable request.].

***Comments from PLOS Editorial Office:** We note that one or more reviewers has recommended that you cite specific previously published works. As always, we recommend that you please review and evaluate the requested works to determine whether they are relevant and should be cited. It is not a requirement to cite these works. We appreciate your attention to this request.*

Reviewers' comments:

Reviewer's Responses to Questions

**Comments to the Author**

1. Is the manuscript technically sound, and do the data support the conclusions?

Reviewer #1: Partly

Reviewer #2: Yes

2. Has the statistical analysis been performed appropriately and rigorously? 

Reviewer #1: No

Reviewer #2: Yes

3. Have the authors made all data underlying the findings in their manuscript fully available?

Reviewer #1: Yes

Reviewer #2: Yes

4. Is the manuscript presented in an intelligible fashion and written in standard English?

Reviewer #1: Yes

Reviewer #2: No

5. Review Comments to the Author

Reviewer #1: 1. Two relevant previous studies are available that are missing from the manuscript:

a) Gupta AK, Sharma R, Sah RP, Sharma S, Jha A, Chapagai M, Saeed F, Shoib S. Cross‐cultural adaptation of Nepalese literacy and stigma of suicide scales (LOSS‐SF‐Nep and SOSS‐SF‐Nep) among Nepalese medical and nursing students. Brain and behavior. 2023 Dec;13(12):e3344.

b) Shah S, Neupane D, Sah KK. Literacy of suicide among doctors and nurses at a tertiary care hospital in Nepal. Journal of College of Medical Sciences-Nepal. 2022 Jun 30;18(2):93-102.

2. LOSS-SF and SOSS-SF instruments seem to used in English language in your study though they have been adapted in Nepali language before. [REF: Gupta AK, Sharma R, Sah RP, Sharma S, Jha A, Chapagai M, Saeed F, Shoib S. Cross‐cultural adaptation of Nepalese literacy and stigma of suicide scales (LOSS‐SF‐Nep and SOSS‐SF‐Nep) among Nepalese medical and nursing students. Brain and behavior. 2023 Dec;13(12):e3344.]

3. The mean score of LOSS-SF-Nep was 6.36 ± 1.92 as per Gupta et al, 2021 which is higher than your findings. Your lower score could possibly be because using the instrument in Nepalese language could bring higher understanding of the questionnaire and accurate literacy.

4. Just like LOSS, SOSS does not seem to be validated or adapted for the study population. It is difficult to understand how the words like shallow. lost, disconnected, etc were communicated or understood by the study volunteers whose mother tongue is not English and more than one-third have studied below 8th standard! This technical error could have been easily avoided by using the validated scale.

5. Your conclusion seem to be partially based on your findings. For example, you say that this study confirms that suicide literacy among Nepalese college students is low, and stigma remains a significant barrier to help-seeking behaviors. I have a question: Stigma is known barrier to help-seeking but how does this study conclude this statement? Similarly, when you say that Demographic variations, such as gender and academic year, further influence levels of stigma and literacy, it is prudent to ask if these variables influence or are associated?

Reviewer #2: 1.what all undergrad courses were included? any exclusion? since under graduate medical student might be more aware of suicide as already included in course curriculum? does it affect study result?

2. study mentions nature of college. any significance of nature of college?

3. consent verbal/written of parents not clearly mentioned?

4. Grammatical errors , typological errors found.

6. PLOS authors have the option to publish the peer review history of their article (what does this mean? ). If published, this will include your full peer review and any attached files.

**Do you want your identity to be public for this peer review?** For information about this choice, including consent withdrawal, please see our Privacy Policy .

Reviewer #1: **Yes: ** Anoop Krishna Gupta

Reviewer #2: No

---

## [Author Response · Author response to Decision Letter 1]

11 May 2025

Dear Editor,

We sincerely thank you and the reviewers for your valuable feedback and the time dedicated to improving our manuscript. We have carefully considered each comment and made the necessary revisions to address the suggestions.

As suggested by the editors, we have explicitly stated that:

Additionally, we have uploaded the data sheet as requested.

Below is a detailed response to each point raised by the reviewers.

Reviewer #1:

1. Two relevant previous studies are available that are missing from the manuscript:

a) Gupta AK, Sharma R, Sah RP, Sharma S, Jha A, Chapagai M, Saeed F, Shoib S. Cross‐cultural adaptation of Nepalese literacy and stigma of suicide scales (LOSS‐SF‐Nep and SOSS‐SF‐Nep) among Nepalese medical and nursing students. Brain and behavior. 2023 Dec;13(12):e3344.

b) Shah S, Neupane D, Sah KK. Literacy of suicide among doctors and nurses at a tertiary care hospital in Nepal. Journal of College of Medical Sciences-Nepal. 2022 Jun 30;18(2):93-102.

Thank you for your valuable time and for suggesting the missing citation. We have included them in our revised discussion section, strengthening the discussion

2. LOSS-SF and SOSS-SF instruments seem to used in English language in your study, though they have been adapted in Nepali language before. [REF: Gupta AK, Sharma R, Sah RP, Sharma S, Jha A, Chapagai M, Saeed F, Shoib S. Cross‐cultural adaptation of Nepalese literacy and stigma of suicide scales (LOSS‐SF‐Nep and SOSS‐SF‐Nep) among Nepalese medical and nursing students. Brain and behavior. 2023 Dec;13(12):e3344.]

Thank you for notifying us about the tool validation. The data for this study were collected during 2022, and at that time, the mentioned validation study was not published. Therefore, our research team consulted with some experts and ensured translation validity. The presence of a validated tool is crucial progress, and we have highlighted the existence of a validated tool in the discussion section. We, however, have not included this validation study in the method section.

3. The mean score of LOSS-SF-Nep was 6.36 ± 1.92 as per Gupta et al, 2021 which is higher than your findings. Your lower score could possibly be because using the instrument in Nepalese language could bring higher understanding of the questionnaire and accurate literacy.

Thank you for your critical observation. We have included this difference in our discussion section and acknowledge the limitation of our study regarding the tools. We have further suggested that future researchers to use the mentioned validated Nepalese version of the tool in the discussion section.

4. Just like LOSS, SOSS does not seem to be validated or adapted for the study population. It is difficult to understand how the words like shallow. lost, disconnected, etc were communicated or understood by the study volunteers whose mother tongue is not English and more than one-third have studied below 8th standard! This technical error could have been easily avoided by using the validated scale.

Thank you for your comment. As mentioned above, at the time of data collection, the validated tool was unavailable, and we had consulted with experts and had also pretested the tool before going into the field to address the issues of misinterpretation and translation validity. But we acknowledge the tool might had brings some reporting bias and have included it as a limitation of our study in light of your valuable feedback.

5. Your conclusion seem to be partially based on your findings. For example, you say that this study confirms that suicide literacy among Nepalese college students is low, and stigma remains a significant barrier to help-seeking behaviors. I have a question: Stigma is known barrier to help-seeking but how does this study conclude this statement? Similarly, when you say that Demographic variations, such as gender and academic year, further influence levels of stigma and literacy, it is prudent to ask if these variables influence or are associated?

Thank you for the suggestion. We have received the conclusion, making it more inline with our findings.

Reviewer #2:

What all undergrad courses were included? any exclusion? since under graduate medical student might be more aware of suicide as already included in course curriculum? does it affect study result?

Thank you for your query. We have covered all six faculties available insight Pokhara Metropolitan for better representation. Considering the word economy we had not mentioned each of the processes, we thank you for helping us clarify this aspect. We had representation from Engineering (including BSc programs in Civil, Electrical, Mechanical, and Computer Engineering programs), Management (including BBA, BBS, and BBA-BI), and Law (BALLB and LLB). Allied Sciences refers to health-related disciplines such as Medicine, Dental, Nursing, Public Health, Pharmacy, physiotherapy, optometry, and Medical Laboratory Technology, optometry. Sciences refers to pure science disciplines, including BSc programs in Chemistry, Physics, Biology, Microbiology, and Forestry. Humanities were also represented through programs such as BDevS, Arts, B.Ed, and Humanities courses.

In this study, the students enrolled in the first year were excluded, as they had just joined their courses and were not regular attendees. There were no exclusions made based on academic course.

In regard to the presence of suicide related courses in undergraduate medical students, certainly it can be assumed that undergraduate students enrolled in medicine, public health, nursing, Pharmacy, etc, have certain sessions related to suicide, and this might be associated with their literacy and/or stigma related to suicide. This is one of the reasons we did not make any exclusions in regards to academic courses, as it allowed us to check if there is any statistically significant difference in stigma and literacy related to suicide across academic disciplines. We have presented this finding in our Table 5 and have further revised the discussion section.

2. Study mentions nature of college. any significance of nature of college?

Thank you for your query. The nature of the college is usually private or public. However, there was no significant difference in stigma and literacy related to suicide across the nature of the college.

3. consent verbal/written of parents not clearly mentioned?

Thank you for notifying this missing information in our sample. only one student was 17 years old, for whom we had acquired parental consent by providing a consent form to the student a few days before data collection. For all students, written informed consent was secured day of data collection before the start of data collection.

4. Grammatical errors , typological errors found.

Thank you !

---

## [Decision Letter · Decision Letter 1]

7 Jun 2025

PONE-D-25-08118R1Suicide Literacy and Stigma among Undergraduate Students in Pokhara, Nepal: A Cross-Sectional StudyPLOS ONE

Dear Dr. Khatri,

Thank you for submitting your manuscript to PLOS ONE. After much discussion, we feel that the manuscript required further clarification mostly in the methodology section as raised by our third reviewer. Therefore, we invite you to submit a revised version of the manuscript that addresses the points raised during the review process.

We look forward to receiving your revised manuscript.

Kind regards,

Saraswati Dhungana, MD

Academic Editor

PLOS ONE

Reviewers' comments:

Reviewer's Responses to Questions

**Comments to the Author**

1. If the authors have adequately addressed your comments raised in a previous round of review and you feel that this manuscript is now acceptable for publication, you may indicate that here to bypass the “Comments to the Author” section, enter your conflict of interest statement in the “Confidential to Editor” section, and submit your "Accept" recommendation.

Reviewer #3: (No Response)

2. Is the manuscript technically sound, and do the data support the conclusions?

Reviewer #3: Partly

3. Has the statistical analysis been performed appropriately and rigorously? 

Reviewer #3: No

4. Have the authors made all data underlying the findings in their manuscript fully available?

Reviewer #3: Yes

5. Is the manuscript presented in an intelligible fashion and written in standard English?

Reviewer #3: Yes

6. Review Comments to the Author

Reviewer #3: The manuscript requires further improvement.

Page 5: The statement ‘to account for potential variability, the sample size was adjusted by applying a design effect of 1.5 times the sample size, resulting in 576 required samples’ requires revision. e.g. ‘A design effect of 1.5 was applied to the sample size to account for clustering in the sampling design.

Page 5: The questionnaires' language version and the validity and reliability information of their translated versions are to be provided. The scoring method for the questionnaires to be provided.

Page 5: A statement/ description on missing values and the method of handling the missing data (if any) is to be mentioned.

Page 6: The citation of the statistical software used and the publisher is to be provided.

A statement on the fulfillment of assumptions for the statistical tests used is to be stated.

Page 6: The purpose of using the specific statistical test is to be clearly mentioned. The outcome variables are to be stated in the statistical analysis section.

Page 6: One or two-tailed test is to be mentioned. The specific name/type of ANOVA is to be stated.

For ANOVA, post hoc analyses are to be presented/highlighted.

Table 1: The overall sample size (N) is to be clearly stated. The variable 'academic year' includes only 595 students instead of 597. If there is missing data, this should be acknowledged and explained.

Table 4: The word agreement is to be denoted e.g. what agreement refers to.

Table 6 title requires revision to be more specific (as compared to Table 5 title).

Page 6: It was mentioned ‘ 95% confidence interval’, but there was no 95% CI presented in the results section.

Tables 5 and 6: Statistical tests are to be denoted in the table footnote.

All statistical tests used in the results are to be mentioned in the statistical analyses in the method section.

The name of the correlation test is to be stated.

Page 11: Table 5 is to be cited in the text.

Page 12: The sentence ‘The comparative analysis revealed that individuals’ requires revision.

Page 13: The paragraph section on correlation requires revision by adapting the words weak and moderate correlation into it where applicable.

Page 13: Related to peer support (r = 0.392) is an incorrect statement. The sentence requires improvement.

Page 16: Correlation value is to be provided where appropriate.

Further statistical analyses e.g. regression, could be performed.

The word p-value in the tables could be replaced with p

In some parts of the discussion, it is unclear whether the points, results, or findings refer to this current study or to other studies. The clarity and coherence of the write-up could be improved.

Some references in the list did not comply with the journal format. e.g. Journal Name

7. PLOS authors have the option to publish the peer review history of their article (what does this mean? ). If published, this will include your full peer review and any attached files.

**Do you want your identity to be public for this peer review?** For information about this choice, including consent withdrawal, please see our Privacy Policy .

Reviewer #3: No

---

## [Author Response · Author response to Decision Letter 2]

25 Jun 2025

Dear Editor,

We sincerely thank you and the reviewers for your valuable feedback and the time dedicated to improving our manuscript. We have carefully considered each comment and made the necessary revisions to address the suggestions.

As suggested by the editors, we have explicitly stated that:

Below is a detailed response to each point raised by the reviewers.

1. Page 5: The statement ‘to account for potential variability, the sample size was adjusted by applying a design effect of 1.5 times the sample size, resulting in 576 required samples’ requires revision. e.g. ‘A design effect of 1.5 was applied to the sample size to account for clustering in the sampling design.

• Response: Thank you for the suggestion. We have revised this section accordingly.

2. Page 5: The questionnaires' language version and the validity and reliability information of their translated versions are to be provided. The scoring method for the questionnaires to be provided.

• Response : Thank you for the comment. The original versions of LOSS, SOSS, and SIDAS were used in the study and we have cited them within the manuscript while mentioning them. At the time this study was implemented, there were no validated Nepalese versions of these tools; thus, we took the help of bilingual mental health experts to translate these tools in Nepali language and also conducted a pretest to ensure translation validity. We have mentioned this in our manuscript. Thank you for your comment, we have added the use Nepali version of the tool in the manuscript.

3. Page 5: A statement/ description on missing values and the method of handling the missing data (if any) is to be mentioned.

• In this study, we had approached 600 students, and 597 students provided a complete response. This was above the minimum required sample size so the data from 597 students were included in this study. We have added this revision in our manuscript.

4. Page 6: The citation of the statistical software used and the publisher is to be provided.

• Response: Thank you for your suggestion. We have cited the statistical software used

5. A statement on the fulfillment of assumptions for the statistical tests used is to be stated.

• Response: Thank you for your suggestion, we have revised the analysis section to clearly state the statistical test used in the study.

6. Page 6: The purpose of using the specific statistical test is to be clearly mentioned. The outcome variables are to be stated in the statistical analysis section.

• Response: The normality of the data was evaluated using the Shapiro-Wilk normality test, a common statistical tool used to assess the normality assumption of a dataset. As the data followed normal distribution independent student t-tests and ANOVA were performed. We have revised the analysis section to clarify this.

7. Page 6: One or two-tailed test is to be mentioned. The specific name/type of ANOVA is to be stated. For ANOVA, post hoc analyses are to be presented/highlighted.

• Response: Statistical significance was set for a two-sided P-value < 0.05. One-way analysis of variance (ANOVA), and post-hoc analysis was performed using the Turkey post-hoc test with Games-Howell.

8. Table 1: The overall sample size (N) is to be clearly stated. The variable 'academic year' includes only 595 students instead of 597. If there is missing data, this should be acknowledged and explained.

• Response: Thank you for notifying the typo error in data distribution of academic year, we have corrected it in the revised manuscript.

9. Table 4: The word agreement is to be denoted e.g. what agreement refers to.

• Response: The word agreement denotes participants' response in SOSS, where they responded agreeing or strongly agreeing with the respective statement.

10. Table 6 title requires revision to be more specific (as compared to Table 5 title).

• Response: Thank you for your suggestion. We have revised it accordingly.

11. Page 6: It was mentioned ‘95% confidence interval’, but there was no 95% CI presented in the results section.

• Response: Thank you for your comment. We have removed the statement regarding 95% confidence interval from the methods section.

12. Tables 5 and 6: Statistical tests are to be denoted in the table footnote.

• Response: Thank you for the suggestion. We have denoted the statistical tests in the table footnote.

13. All statistical tests used in the results are to be mentioned in the statistical analyses in the method section.

• Response: We have mentioned all the statistical test used in the method section.

14. The name of the correlation test is to be stated.

• Response: Thank you for your suggestion. The name of the correlation test has been mention in the manuscript

15. Page 11: Table 5 is to be cited in the text.

• Response: Thank you for notifying this missing citation. We have cited a table in the text.

16. Page 12: The sentence ‘The comparative analysis revealed that individuals’ requires revision.

• Response: Thank you for your suggestion. We have revised it accordingly.

17. Page 13: The paragraph section on correlation requires revision by adapting the words weak and moderate correlation into it where applicable.

• Response: Thank you for your valuable suggestion. We have revised the paragraph section on correlation by incorporating the terms weak, moderate, and high correlation where applicable.

18. Page 13: Related to peer support (r = 0.392) is an incorrect statement. The sentence requires improvement.

• Response: Thank you for the suggestion. We have revised the statements.

19. Page 16: Correlation value is to be provided where appropriate.

• Response : Thank you for your suggestion. We have included the correlation values where appropriate

20. Further statistical analyses e.g. regression, could be performed.

• Response: Thank you for your valuable suggestion regarding the inclusion of regression analysis. We acknowledge that regression analysis could provide insights into predictive relationships. However, our primary objective was to explore and compare suicide literacy and stigma levels across different socio-demographic and psychosocial groups. The study was exploratory in nature, and we used appropriate comparative and correlational analyses (e.g., t-tests, ANOVA, and Pearson correlation) to meet this objective. We respectfully chose not to perform regression analysis in this manuscript and we thank you for your valuable suggestion and kind consideration.

21. The word p-value in the tables could be replaced with p

• Response: Thank you for your suggestion. We have revised accordingly.

---

## [Decision Letter · Decision Letter 2]

17 Jul 2025

PONE-D-25-08118R2Suicide Literacy and Stigma among Undergraduate Students in Pokhara, Nepal: A Cross-Sectional StudyPLOS ONE

Dear Dr. Khatri,

Thank you for submitting your manuscript to PLOS ONE. After careful consideration, we feel that it has merit but does not fully meet PLOS ONE’s publication criteria as it currently stands. Therefore, we invite you to submit a revised version of the manuscript that addresses the points raised during the review process. Reviewer 3 has some minor comments as follows. Please respond to the following before we can proceed further. 

The authors’ justification for not performing regression analysis, stating that "our primary objective was to explore and compare suicide literacy and stigma levels across different socio-demographic and psychosocial groups " does not align well with the stated objective of the study: "This study examines suicide literacy and stigma among undergraduate students and its associated factors.' This discrepancy should be addressed, and the objective should be revised for clarity and consistency. The discussion needs to be revisited due to the exploratory nature of the study, as stated by the authors.

We look forward to receiving your revised manuscript.

Kind regards,

Saraswati Dhungana, MD

Academic Editor

PLOS ONE

Journal Requirements:

Reviewers' comments:

Reviewer's Responses to Questions

**Comments to the Author**

1. If the authors have adequately addressed your comments raised in a previous round of review and you feel that this manuscript is now acceptable for publication, you may indicate that here to bypass the “Comments to the Author” section, enter your conflict of interest statement in the “Confidential to Editor” section, and submit your "Accept" recommendation.

Reviewer #3: (No Response)

2. Is the manuscript technically sound, and do the data support the conclusions?

Reviewer #3: Partly

3. Has the statistical analysis been performed appropriately and rigorously? 

Reviewer #3: No

4. Have the authors made all data underlying the findings in their manuscript fully available?

Reviewer #3: Yes

5. Is the manuscript presented in an intelligible fashion and written in standard English?

Reviewer #3: Yes

6. Review Comments to the Author

Reviewer #3: The authors’ justification for not performing regression analysis, stating that "our primary objective was to explore and compare suicide literacy and stigma levels across different socio-demographic and psychosocial groups " does not align well with the stated objective of the study: "This study examines suicide literacy and stigma among undergraduate students and its associated factors.' This discrepancy should be addressed, and the objective should be revised for clarity and consistency. The discussion needs to be revisited due to the exploratory nature of the study, as stated by the authors.

7. PLOS authors have the option to publish the peer review history of their article (what does this mean? ). If published, this will include your full peer review and any attached files.

**Do you want your identity to be public for this peer review?** For information about this choice, including consent withdrawal, please see our Privacy Policy .

Reviewer #3: No

---

## [Author Response · Author response to Decision Letter 3]

5 Aug 2025

Dear Editor,

We sincerely thank you and the reviewers for your valuable feedback and the time dedicated to improving our manuscript. We have carefully considered each comment and made the necessary revisions to address the suggestions.

As suggested by the editors, we have explicitly stated that:

Below is a detailed response to point raised by the reviewers.

The authors’ justification for not performing regression analysis, stating that "our primary objective was to explore and compare suicide literacy and stigma levels across different socio-demographic and psychosocial groups " does not align well with the stated objective of the study: "This study examines suicide literacy and stigma among undergraduate students and its associated factors.' This discrepancy should be addressed, and the objective should be revised for clarity and consistency. The discussion needs to be revisited due to the exploratory nature of the study, as stated by the authors.

Response: Thank you for your insightful comment. We have performed multiple linear regression analyses to identify factors associated with suicide stigma and literacy. The results of these analyses, including full coefficient tables for all models (SOSS-Stigma, SOSS-Depression, SOSS-Glorification, and LOSS-SF), are provided in Supplementary File S1.

Thank you!

---

## [Editor Report · Decision Letter 3]

13 Aug 2025

Suicide Literacy and Stigma among Undergraduate Students in Pokhara, Nepal: A Cross-Sectional Study

PONE-D-25-08118R3

Dear Dr. Khatri,

We’re pleased to inform you that your manuscript has been judged scientifically suitable for publication and will be formally accepted for publication once it meets all outstanding technical requirements.

Kind regards,

Saraswati Dhungana, MD

Academic Editor

PLOS ONE

---

## [Editor Report · Acceptance letter]

PONE-D-25-08118R3

PLOS ONE

Dear Dr. Khatri,

I'm pleased to inform you that your manuscript has been deemed suitable for publication in PLOS ONE. Congratulations! Your manuscript is now being handed over to our production team.

Kind regards,

on behalf of

Dr. Saraswati Dhungana

Academic Editor

PLOS ONE